# EMERGENT WORLD REPRESENTATIONS: EXPLORING A SEQUENCE MODEL TRAINED ON A SYNTHETIC TASK

**Kenneth Li**[*]
Harvard University

**Aspen K. Hopkins**
Massachusetts Institute of Technology

**David Bau**
Northeastern University

**Fernanda Viégas**
Harvard University

**Hanspeter Pfister**
Harvard University

**Martin Wattenberg**
Harvard University

## ABSTRACT

Language models show a surprising range of capabilities, but the source of their apparent competence is unclear. Do these networks just memorize a collection of surface statistics, or do they rely on internal representations of the process that generates the sequences they see? We investigate this question in a synthetic setting by applying a variant of the GPT model to the task of predicting legal moves in a simple board game, Othello. Although the network has no a priori knowledge of the game or its rules, we uncover evidence of an emergent nonlinear internal representation of the board state. Interventional experiments indicate this representation can be used to control the output of the network. By leveraging these intervention techniques, we produce "latent saliency maps" that help explain predictions. [1]

## 1 INTRODUCTION

Recent language models have shown an intriguing range of capabilities. Networks trained on a simple "next-word" prediction task are apparently capable of many other things, such as solving logic puzzles or writing basic code. [2] Yet how this type of performance emerges from sequence predictions remains a subject of current debate.

Some have suggested that training on a sequence modeling task is inherently limiting. The arguments range from philosophical (Bender & Koller, 2020) to mathematical (Merrill et al., 2021). A common theme is that seemingly good performance might result from memorizing "surface statistics," i.e., a long list of correlations that do not reflect a causal model of the process generating the sequence. This issue is of practical concern, since relying on spurious correlations may lead to problems on out-of-distribution data (Bender et al., 2021; Floridi & Chiriatti, 2020).

On the other hand, some tantalizing clues suggest language models may do more than collect spurious correlations, instead building interpretable *world models*—that is, understandable models of the process producing the sequences they are trained on. Recent evidence suggests language models can develop internal representations for very simple concepts, such as color, direction Abdou et al. (2021); Patel & Pavlick (2022), or tracking boolean states during synthetic tasks (Li et al., 2021) (see Related Work (section 6) for more detail).

A promising approach to studying the emergence of world models is used by Toshniwal et al. (2021), which explores language models trained on chess move sequences. The idea is to analyze the behavior of a standard language modeling architecture in a well-understood, constrained setting. The paper finds that these models learn to predict legal chess moves with high accuracy. Furthermore, by analyzing predicted moves, the paper shows that the model appears to track the board state. The authors stop short, however, of exploring the form of any internal representations. Such an

---

[*]Correspondence to `ke_li@g.harvard.edu`
[1]Codes at `https://github.com/likenneth/othello_world`
[2]See Srivastava et al. (2022) for an encyclopedic list of examples.

investigation will be the focus of this paper. A key motivation is the hope that ideas and techniques learned in this simplified setting may eventually be useful in natural-language settings as well.

## 1.1 The game of Othello as testbed for interpretability

Toshniwal et al. (2021)'s observations suggest a new approach to studying the representations learned by sequence models. If we think of a board as the "world," then games provide us with an appealing experimental testbed to explore world representations of moderate complexity. As our setting, we choose the popular game of Othello (Figure 1), which is simpler than chess. This setting allows us to investigate world representations in a highly controlled context, where both the task and sequence being modeled are synthetic and well-understood.

As a first step, we train a language model (a GPT variant we call Othello-GPT) to extend partial game transcripts (a list of moves made by players) with legal moves. **The model has no a priori knowledge of the game or its rules**. All it sees during training is a series of tokens derived from the game transcripts. Each token represents a tile where players place their discs. Note that we do *not* explicitly train the model to make strategically good moves or to win the game. Nonetheless, our model is able to generate legal Othello moves with high accuracy.

Our next step is to look for world representations that might be used by the network. In Othello, the "world" consists of the current board position. A natural question is whether, within the model, we can identify a representation of the board state involved in producing its next move predictions. To study this question, we train a set of probes, i.e., classifiers which allow us to infer the board state from the internal network activations . This type of probing has become a standard tool for analyzing neural networks (Alain & Bengio, 2016; Tenney et al., 2019; Belinkov, 2016).

Using this probing methodology, we find evidence for an emergent world representation. In particular, we show that a non-linear probe is able to predict the board state with high accuracy (section 3). (Linear probes, however, produce poor results.) This probe defines an internal representation of the board state. We then provide evidence that this representation plays a causal role in the network's predictions. Our main tool is an intervention technique that modifies internal activations so that they correspond to counterfactual board states.

We also discuss how knowledge of the internal world model can be used as an interpretability tool. Using our activation-intervention technique, we create *latent saliency maps*, which provide insight into how the network makes a given prediction. These maps are built by performing attribution at a high-level setting (the board) rather than a low-level one (individual input tokens or moves).

To sum up, we present four contributions: (1) we provide evidence for an emergent world model in a GPT variant trained to produce legal moves in Othello; (2) we compare the performance of linear and non-linear probing approaches, and find that non-linear probes are superior in this context; (3), we present an intervention technique that suggests that, in certain situations, the emergent world model can be used to control the network's behavior; and (4) we show how probes can be used to produce *latent saliency maps* to shed light on the model's predictions.

## 2 "Language modeling" of Othello game transcripts

Our approach for investigating internal representations of language models is to narrow our focus from natural language to a more controlled synthetic setting. We are partly inspired by the fact that language models show evidence of learning to make valid chess moves simply by observing game transcripts in training data (Toshniwal et al., 2021). We choose the game Othello, which is simpler than chess, but maintains a sufficiently large game tree to avoid memorization. Our strategy is to see what, if anything, a GPT variant learns simply by observing game transcripts, without any a priori knowledge of rules or board structure.

The game is played on an 8x8 board where two players alternate placing white or black discs on the board tiles. The object of the game is to have the majority of one's color discs on the board at the end of the game. Othello makes a natural testbed for studying emergent world representations since the game tree is far too large to memorize, but the rules and state are significantly simpler than chess.

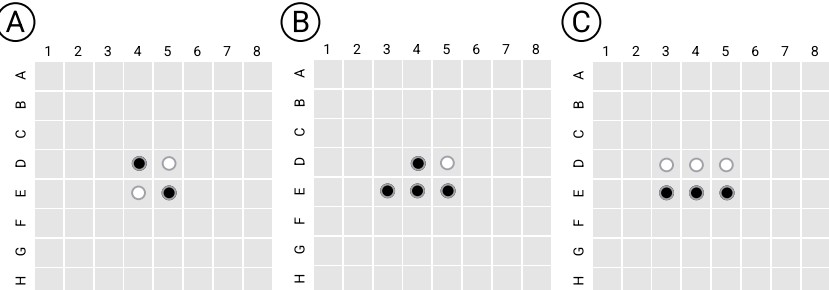

Figure 1: A visual explanation of Othello rules, from left to right: (A) The board is always initialized with four discs (two black, two white) placed in the center of the board. (B) Black always moves first. Every move must flip one or more opponent discs by outflanking—or sandwiching—the opponent disc(s). (C) The opponent repeats this process. A game ends when there are no more legal moves.

The following subsections describe how we train a system with no prior knowledge of Othello to predict legal moves with high accuracy. The system itself is not our end goal; instead, it serves as our object of study.

## 2.1 DATASETS: "CHAMPIONSHIP" AND "SYNTHETIC"

We use two sets of training data for the system, which we call "championship" and "synthetic". Each captures different objectives, namely data quality vs. quantity. While limited in size, championship data reflects strategic moves by expert human players. The synthetic data set is far larger, consisting of legal but otherwise random moves.

Our *championship dataset* is produced by collecting Othello championship games from two online sources[3], containing $7,605$ and $132,921$ games, respectively. They are combined and split randomly by $8:2$ into training and validation sets. The games in this dataset were produced by matches where human players presumably made moves with a strategic intent to win. Following this, we generate a *synthetic dataset* with 20 million games for training and $3,796,010$ games for validation. We compute this dataset by uniformly sampling leaves from the Othello game tree. Its data distribution is different from the championship games, reflecting no strategy.

## 2.2 MODEL AND TRAINING

Our goal is to study how much Othello-GPT can learn from pure sequence information, so we provide as few inductive biases as possible. (Note the contrast with a system like AlphaZero (Silver et al., 2018), where the goal was to win highly competitive chess games.) We therefore use only sequential tile indices as input to our model. For example, A4 and H6 are indexed as the 4th and the 58th word in our vocabulary, respectively. Each game is treated as a sentence tokenized with a vocabulary of 60 words, where each word corresponds to one of the 60 tiles on which players put discs, excluding the 4 tiles in the center (Figure 1).

We trained an 8-layer GPT model (Radford et al., 2018; 2019; Brown et al., 2020) with an 8-head attention mechanism and a 512-dimensional hidden space. The training was performed in an autoregressive fashion. For each partial game $\{y_t\}_{t=0}^{T-1}$, the computation process starts from indexing a trainable word embedding consisting of the 60 vectors, each for one word, to get $\{x_t^0\}_{t=0}^{T-1}$. They are then sequentially processed by 8 multi-head attention layers. We denote the intermediate feature for the $t$-th token after the $l$-th layer as $x_t^l$. By employing a causal mask, only the features at the immediately preceding layer and earlier time steps $x_{\leq t}^{l-1}$ are visible to $x_t^l$. Finally, $x_{T-1}^8$ goes through a linear classifier to predict logits for $\hat{y}_T$. We minimize the cross-entropy loss between ground-truth move and predicted logits by gradient descent.

The model starts from randomly initialized weights, including in the word embedding layer. Though there are geometrical relationships between the 60 words (e.g., C4 is below B4), this inductive bias is not explicitly given to the model but rather left to be learned.

---

[3]`www.liveothello.com` and `www.ffothello.org`.

## 2.3 OTHELLO-GPT USUALLY PREDICTS LEGAL MOVES

We now evaluate how well the model's predictions adhere to the rules of Othello. For each game in the validation set, which was not seen during training, and for each step in the game, we ask Othello-GPT to predict the next legal move conditioned by the partial game before that move. We then calculate the error rate by checking if the top-1 prediction is legal. The error rate is $0.01\%$ for Othello-GPT trained on the synthetic dataset and $5.17\%$ for Othello-GPT trained on the championship dataset. For comparison, the untrained Othello-GPT has an error rate of $93.29\%$. The main takeaway is that **Othello-GPT does far better than chance in predicting legal moves** when trained on both datasets.

A potential explanation for these results may be that Othello-GPT is simply memorizing all possible transcripts. To test for this possibility, we created a *skewed dataset* of 20 million games to replace the training set of synthetic dataset. At the beginning of every game, there are four possible opening moves: C5, D6, E3 and F4. This means the lowest layer of the game tree (first move) has four nodes (the four possible opening moves). For our skewed dataset, we truncate one of these nodes (C5), which is equivalent to removing a quarter of the whole game tree. Othello-GPT trained on the skewed dataset still yields an error rate of $0.02\%$. Since Othello-GPT has seen none of these test sequences before, pure sequence memorization cannot explain its performance [4].

If the performance of Othello-GPT is not due to memorization, what is it doing? We now turn to this question by probing for internal representations of the game state.

## 3 EXPLORING INTERNAL REPRESENTATIONS WITH PROBES

We seek to understand if Othello-GPT computes internal representations of the game state. One standard tool for this task is a "probe" (Alain & Bengio, 2016; Belinkov, 2016; Tenney et al., 2019). A probe is a classifier or regressor whose input consists of internal activations of a network, and which is trained to predict a feature of interest, e.g., part of speech or parse tree depth (Hewitt & Manning, 2019). If we are able to train an accurate probe, it suggests that a representation of the feature is encoded in the network's activations.

In our case, we want to know whether Othello-GPT's internal activations contain a representation of the current board state. To study this question, we train probes that predict the board state from the network's internal activations after a given sequence of moves. Note that the board state—whether each tile is empty or holds a black or white disc—is generally a nonlinear function of the input tokens. On the other hand, it is straightforward to write a program to compute this function, it makes a natural probe target.[5]

We take the autoregressive features $x_t^l$ that summarize the partial sequence $y_{\leqslant t}$ as the input to the probe and study results from different layers $l$. The output $p_\theta(x_t^l)$ is a 3-way categorical probability distribution. We randomly split pairs of internal representation and ground-truth tile state by $8:2$ into training and validation set. Error rates on validation set are reported. A best random guess yields $52.95\%$, if the probe always guess the tile is empty.

### 3.1 LINEAR PROBES HAVE HIGH ERROR RATES

Our first result is that linear classifier probes have poor relative accuracy. Its function can be written as $p_\theta(x_t^l) = \mathrm{softmax}(W x_t^l)$ where $\theta = \{W \in \mathbb{R}^{F \times 3}\}$. $F$ is the number of dimensions of input $x_t^l$. As Table 1 shows, error rates never dip below $20\%$. As a baseline, we have included probes trained on a randomly initialized network[6] We can see that there is only a marginal improvement in accuracy when we move to probing a fully-trained network. This result suggests that if there is an internal representation of the board state, it does not have a simple linear form.

---

[4]Note that even truncated the game tree may include some board states in the test dataset, since different move sequences can lead to the same board state. However, our goal is to prevent memorization of *input data*; the network only sees moves, and never sees board state directly.

[5]Classifying a tile as unoccupied or occupied can be written as a linear function of the input sequence, thus we consider only the 3-way black/white/empty classifiers.

[6]Probes on the randomized network do better than chance; a constant guess of "empty" has a 47% error rate. But that performance is not a surprise, since it makes sense that some information about moves is preserved even by a random network. The key comparison is between the randomized network and the trained network.

|  | $x^1$ | $x^2$ | $x^3$ | $x^4$ | $x^5$ | $x^6$ | $x^7$ | $x^8$ |
|---|---|---|---|---|---|---|---|---|
| Randomized | 26.7 | 27.1 | 27.6 | 28.0 | 28.3 | 28.5 | 28.7 | 28.9 |
| Championship | 24.2 | 23.8 | 23.7 | 23.6 | 23.6 | 23.7 | 23.8 | 24.3 |
| Synthetic | 21.9 | 20.5 | 20.4 | 20.6 | 21.1 | 21.6 | 22.2 | 23.1 |

Table 1: Error rates (%) of linear probes on randomized Othello-GPT and Othello-GPTs trained on different datasets across different layers ($x^i$ represents internal representations after the $i$-th layer).

## 3.2 Nonlinear Probes Have Lower Error Rates

Given the poor performance of linear probes, it is natural to ask whether a nonlinear probe would have higher accuracy. Moving up one notch of complexity, we apply a 2-layer MLP as a probe. This technique has been used successfully in other language model probing work, e.g., Conneau et al. (2018); Cao et al. (2021); Hernandez & Andreas (2021). Its function can be written as $p_\theta(x_t^l) = \text{softmax}(W_1 \text{ReLU}(W_2 x_t^l))$ where $\theta = \{W_1 \in \mathbb{R}^{H \times 3}, W_2 \in \mathbb{R}^{F \times H}\}$. $H$ is the number of hidden dimensions for the nonlinear probes.

The probe accuracy for trained networks, shown in Table 2, is significantly better than the linear probe in absolute terms. By contrast, the baseline (probing a randomized network with nonlinear probes) shows almost no improvement over the linear case. These results indicate that the probe may be recovering a nontrivial representation of board state in the network's activations. In section 4, we describe intervention experiments validating this hypothesis.

|  | $x^1$ | $x^2$ | $x^3$ | $x^4$ | $x^5$ | $x^6$ | $x^7$ | $x^8$ |
|---|---|---|---|---|---|---|---|---|
| Randomized | 25.5 | 25.4 | 25.5 | 25.8 | 26.0 | 26.2 | 26.2 | 26.4 |
| Championship | 12.8 | 10.3 | 9.5 | 9.4 | 9.8 | 10.5 | 11.4 | 12.4 |
| Synthetic | 11.3 | 7.5 | 4.8 | 3.4 | 2.4 | 1.8 | 1.7 | 4.6 |

Table 2: Error rates (%) of nonlinear probes on randomized Othello-GPT and Othello-GPTs trained on different datasets across different layers. Standard deviations are reported in Appendix H.

## 4 Validating Probes with Interventional Experiments

Our nonlinear probe accuracies suggest that Othello-GPT computes information reflecting the board state. It's not obvious, however, whether that information is causal for the model's predictions. In the following section, we adhere to Belinkov (2016)'s recommendation, performing a set of interventional experiments to determine the causal relationship between model predictions and the emergent world representations.

To determine whether the board state information affects the network's predictions, we influence internal activations *during* Othello-GPT's calculation and measure the resulting effects. At a high level, the interventions are as follows: given a set of activations from the Othello-GPT, a probe predicts a baseline board state $B$. We record the move predictions associated with $B$, then modify these activations such that our probe reports an updated board state $B'$. Through our protocol, only a single tile $s$ distinguishes $B'$ from $B$'s board state (an example of which can be seen in Figure 2). This small modification results in a different set of possible legal moves for $B'$. If the new predictions match our expectations for $B'$—and not the predictions we recorded for $B$—we conclude the representation had a causal effect on the model.

### 4.1 Intervention Technique

To implement an intervention that changes the predicted state from a board position $B$ to a modified version $B'$ we must decide (a) which layers to modify activations in, and (b) how to modify those activations. The first question is subtle. Given the causal attention mechanism of GPT-2, modifying activations for only one layer is unlikely to be effective as later layer computations incorporate information from prior board representations unaffected by our intervention. Instead, we select an initial layer $L_s$ then modify it and subsequent layers' activations (see Figure 2 (C)). Our modification uses a simple gradient descent method on the probe's class score for the particular tile $s$ whose state is being modified.

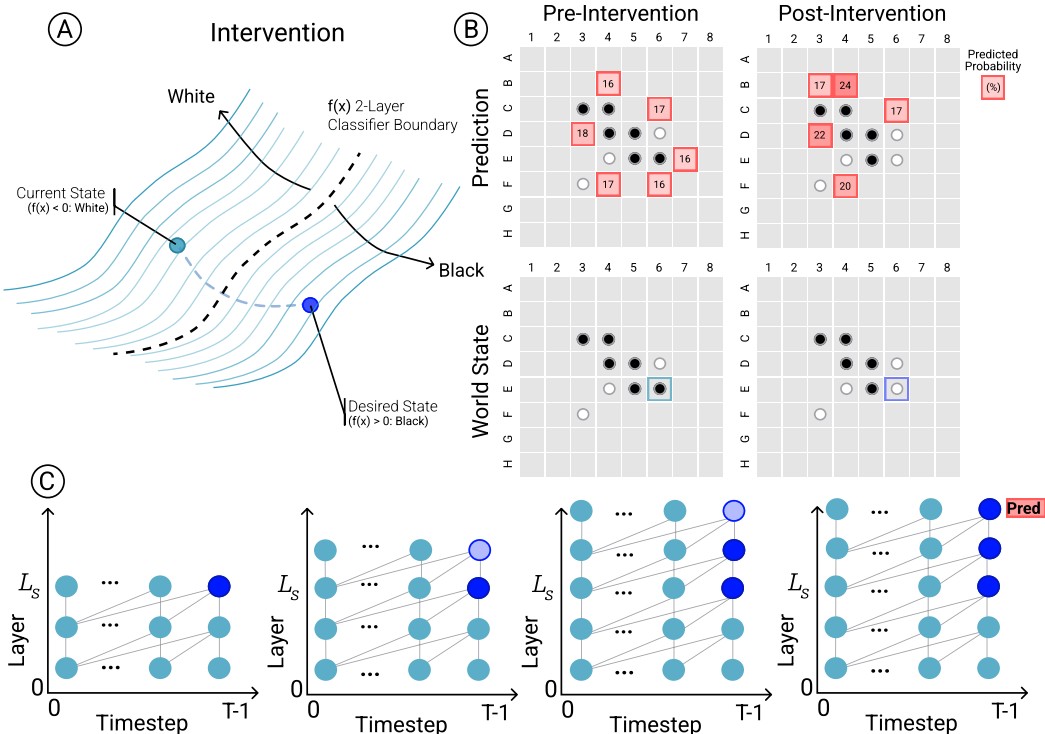

Figure 2: **(A)** explains how we intervene on a board tile. Here, we only want to flip one tile, e.g. E6, from white to black. In **(B)**, four views present an Othello game in progression, which can be reliably probed from an internal representation $x$. The lower left board represents the model's perceived world state prior to intervention. The upper left board shows the model's predictions for legal moves given this state. Post-intervention, the model's world state is updated—E6's state has been switched from white to black (lower right), leading to a different set of legal move predictions (upper right). Note that two tiles (E6) are highlighted in the world state boards. This is the tile that we "intervene" on, changing from white to black. **(C)** Shows our proposed intervention scheme. Light blue indicates unmodified activations; dark blue represents activations affected by intervention. Starting from a predefined layer, we intervene at the temporally-last token (shown in **(A)**). We replace original internal representations with the post-intervention one and resume computation for the next one layer. Part of the misinformation gets corrected (light blue), but we alternate this intervening and computation process until the last layer, from which the next-step prediction is made.

Figure 2 illustrates an intervention on a single feature $x$ into $x'$ such that the corresponding board state $B$ is updated to match the desired $B'$. We observe the effectiveness of these interventions by probing the intervened $x'$ or $x$ at later layers (see Appendix D), as well as the change in next-step prediction (see subsection 4.2). Consistent with the training process of probes $p_\theta$, we use cross entropy loss between the probe-predicted probability distribution and the desired board state, but rather than optimize probe weights $\theta$, we optimize $x$ for intervention [7]:

$$ x' \leftarrow x - \alpha \frac{\partial \mathcal{L}_{\text{CE}}(p_\theta(x), B')}{\partial x}. $$

At timestep $T$, the internal activations of an $L$-layer Othello-GPT can be viewed as an $L \times T$ grid of activation vectors. Our intervention process runs Othello-GPT sequentially, but uses gradient descent to modify key activation vectors at the last timestep such that their board state class scores change. Note that if we change activations only at a middle layer, activations at higher layers are directly affected by pre-intervention information. Thus, we sequentially intervene $\{x^l_{T-1}\}^L_{l=L_s}$ at the last timestep, on all activations starting from a preset layer $L_s$ until the final layer, illustrated in Figure 2.

---

[7]Note that $\alpha$ is the learning rate. See more on the intervention hyper-parameters in Appendix G.

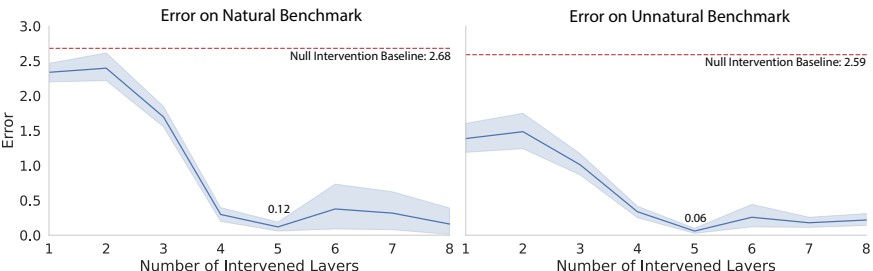

Figure 3: Intervention experiment results. Red dashed lines represents average number of errors by testing pre-intervention predictions on post-intervention ground-truths, representing a null intervention method for contrast. The shaded area represents the $95\%$ confidence interval.

## 4.2 EVIDENCE FOR A CAUSAL ROLE FOR THE REPRESENTATION

To systematically determine if this world representation is causal for model predictions, we create an evaluation benchmark set. A test case in this set consists of a triple of a partial game, a targeted board tile, and a target state. For each case, we give the partial game to Othello-GPT and perform the intervention described in the previous section. I.e., we extract the model's activations mid-computation, modify them to change the representation of the targeted board tile into the target state, give back the modified world representation and let it make a prediction with this new world state.

The benchmark set consists of two subsets of 1000 intervention cases: one "natural," one "unnatural." The natural subset consists of positions reachable by legal play. The unnatural subset contains position that are unreachable by legal play. This second subset is designed to be a stringent test, since it is by definition far from anything encountered in the training distribution.

We measure prediction alignment with the intervened world representation as a multi-label classification problem, comparing the top-$N$ predictions against the ground-truth legal next-move set, where $N$ is the number of legal next-moves after intervention. We then calculate an error per case (a sum of false positives and false negatives, shown in Figure 3)[8]. For both benchmarks, $L_s = 4$ (intervening 5 layers) gives the best result: average errors of **0.12** and **0.06** respectively. Compared to baseline errors (**2.68** and **2.59**), the proposed intervention technique is effective even under unnatural board states, suggesting the emergent representations are causal to model predictions.

## 5 LATENT SALIENCY MAPS: ATTRIBUTION VIA INTERVENTION

The intervention technique of the previous section provides insight into the predictions of Othello-GPT. We can also use it to create visualizations which contextualize Othello-GPT's predictions in terms of the board state. The basic idea is simple. For each tile $s$ on the board $B$, we ask how much the network's prediction probability for the attributed tile p will change if we apply the intervention technique in section 4 to change the state representation of that tile $s$. This will yield a value per tile $s \in B$, positive or negative, corresponding to its saliency in the prediction of p (see algorithm 1). We then create a visualization of the board where tiles are colored according to their saliency for the top-1 prediction for the current board state[9]. Because this map is based on the network's latent space rather than its input, we call it a **latent saliency map**.

Figure 4 shows latent saliency maps for the top-1 predictions for Othello-GPTs trained on the synthetic and championship datasets by intervening from $L_s = 4$. The two diagrams show a clear pattern. The synthetic Othello-GPT shows high saliency for precisely those tiles that are required to make a move legal. In almost all cases, other tiles have lower saliency values. Even without knowing how synthetic-GPT was trained, an experienced Othello player might be able to guess its goal. The latent saliency maps for the championship version, however, are more complex. Although tiles that relate directly to legality typically have high values, many other tiles show high saliency as well. This pattern makes sense, too. Expert moves rely on complex global features of the board. The difference

---

[8]Qualitative results can be found in Appendix C. We report additional metrics in Appendix F.

[9]However, ideally we can attribute any prediction. See Appendix E for more discussion.

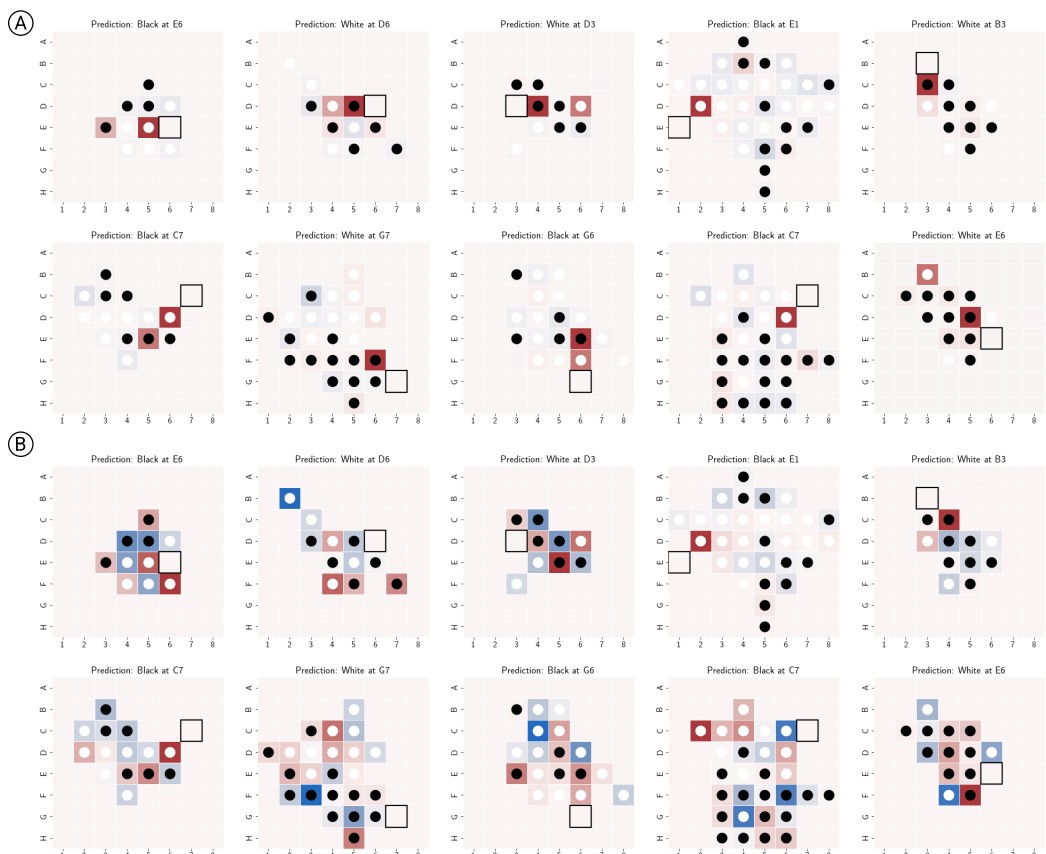

Figure 4: Latent saliency maps: Each subplot shows a different game state, and the top-1 prediction by the model is enclosed in a black box. Colors (red is high, blue is low) indicate the contribution of a square's state to this prediction. The contribution is higher when changing the internal representation of this square makes the prediction less likely. The values are normalized by subtracting the mean of the board. **(A)** Latent saliency maps for Othello-GPT trained on the synthetic dataset, where the model learns *legal* moves. **(B)** Latent saliency maps for Othello-GPT trained on the championship dataset. Rather than learning rules, this Othello-GPT learns to make *strategically good* moves.

between the latent saliency maps for the two versions of Othello-GPT suggests that the visualization technique is providing useful information about the two systems.

---

**Algorithm 1:** Attribution via Intervention on Othello-GPT

**Inputs :**
  $B$    the current board state
  p    a legal next move which we try to attribute
**Output :**
  $\{S_s\}_{s \in B}$    assigned sensitivity values for the prediction of p
  $p_0 \leftarrow f_{\mathrm{p}}(x_{t-1})$        // pre-intervention next-step probability for p
**for** $s \in B$ **do**
  | $\tilde{x}_{t-1} \leftarrow \mathrm{Intervention}(x_{t-1}, s)$
  | $p_s \leftarrow f_{\mathrm{p}}(\tilde{x}_{t-1})$
  | $S_s \leftarrow p_0 - p_s$

---

## 6 RELATED WORK

Our work fits into a general line of investigation into world representations created by sequence models. For example, (Li et al., 2021) fine-tune two language models on synthetic natural language

tasks (Long et al., 2016) and find evidence that semantic information about the underlying world state is at least weakly encoded in the activations of the network. More direct evidence of a faithful representation of 3D color space comes from Abdou et al. (2021), who examine activations in the BERT model and find a geometric connection to a standard 3D color space. Another study by (Patel & Pavlick, 2022) shows that language models can learn to map conceptual domains, e.g., direction and color, onto a grounded world representation via prompting techniques (Brown et al., 2020). These investigations operate in natural language domains, but investigate relatively simple world models.

Another related stream of work concerns neural networks that learn board games. There is a long history of work in AI to learn game moves, but in general, these systems have been given some a priori knowledge of the structure of the game. Even one of the most general-purpose game-playing engines, AlphaZero (Silver et al., 2018), has built-in knowledge of basic board structure and game rules (although, intriguingly, it seems to develop interpretable models of various strategic concepts (McGrath et al., 2021; Forde et al., 2022)). Closer to the work described here–and a major motivation for our research–is Toshniwal et al. (2021) which trains a language model on chess transcripts. They show strong evidence that transformer networks are building a representation of internal board state, but they stop short at investigating what form that representation takes.

The intervention technique of section 4 follows an approach of steering model output while keeping the model frozen. It is related to the ideas behind plug-and-play controllable text generation for autoregressive (Dathathri et al., 2019; Qin et al., 2020; Krause et al., 2020) and diffusion (Li et al., 2022) language models by optimizing the likelihood of the desired attribute and the fluency of generated texts at the same time. These methods naturally involve a trade-off and require several forward and backward passes to generate. Our proposed intervention method stands out by only working on internal representations and requires only one forward pass. Finally, latent saliency maps can be viewed as a generalization of the TCAV approach Kim et al. (2018); Ghorbani et al. (2019); Koh et al. (2020). In the TCAV setting, attribution is performed via directional derivatives. This is essentially a linearization of the gradient-descent optimization used in our attribution maps.

# 7 CONCLUSION

Our experiments provide evidence that Othello-GPT maintains a representation of game board states—that is, the Othello "world"—to produce sequences it was trained on. This representation appears to be nonlinear in an essential way. Further, we find that these representations can be causally linked to how the model makes its predictions. Understanding of the internal representations of a sequence model is interesting in its own right, but may also be helpful in deeper interpretations of the network.

We have also described how interventional experiments may be used to create a "latent saliency map", which gives a picture, in terms of the Othello board, of how the network has made a prediction. Applied to two versions of Othello-GPT that were trained on different data sets, the latent saliency maps highlight the dramatic differences between underlying representations of the Othello-GPT trained on synthetic dataset and its counterpart trained on championship dataset.

There are several potential lines of future work. One natural extension would be to perform the same type of investigations with other, more complex games. It would also be interesting to compare the strategies learned by a sequence model trained on game transcripts with those of a model trained with a priori knowledge of Othello. One option is to compare latent saliency maps of Othello–GPT with standard saliency maps of an Othello-playing program which has the actual board state as input.

More broadly, it would be interesting to study how our results generalize to models trained on natural language. One stepping stone might be to look at language models whose training data has included game transcripts. Will we see similar representation of board state? Grammar engineering tools (Weston et al., 2015; Hermann et al., 2017; Côté et al., 2018) could help define a synthetic data generation process that maps world representations onto natural language sentences, providing a similarly controllable setting like Othello while closing the distance to natural languages. For more complex natural language tasks, can we find meaningful world representations? Our hope is that the tools described in this paper—nonlinear probes, layerwise interventions, and latent saliency maps—may prove useful in natural language settings.

ACKNOWLEDGMENTS

We thank members of the Visual Computing Group and the Insight + Interaction Lab at Harvard for their early discussions and feedback. We especially thank Aoyu Wu for helping with making part of the figures and other valuable suggestions. We gratefully acknowledge the support of Harvard SEAS Fellowship (to KL), Siebel Fellowship (to AH), Open Philanthropy (to DB), and FTX Future Fund Regrant Program (to DB). This work was partially supported by NSF grant IIS-1901030.

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

## A    VISUALIZING THE GEOMETRY OF PROBES

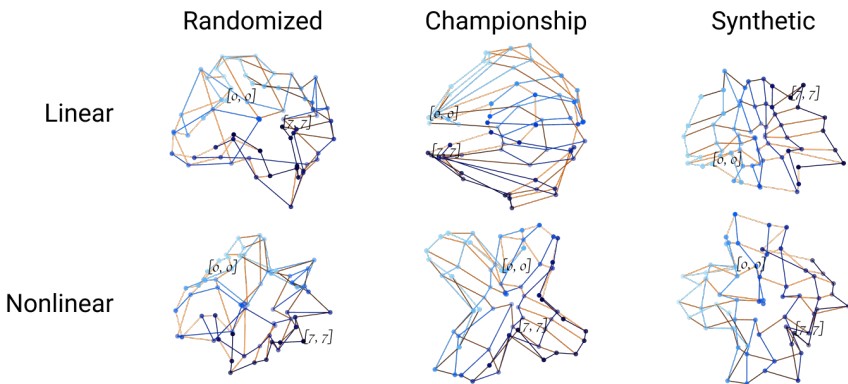

Figure 5: Geometry of probe weights. The first row is for linear probe weights; the second row is for the second-layer weights of the nonlinear probes. Three columns from left to right mean: probes trained on randomly-initialized GPT, on GPTs trained on championship and synthetic datasets respectively. For each subplot, one dot represents one tile on an Othello board, and neighbouring tiles have their two dots connected. Top-left and right-bottom corners are labelled for reference. Togglable 3D plots can be found in Supplementary Material.

Both linear and nonlinear probes can be viewed as geometric objects. In the case of linear probes, we can associate each classifier with the normal vector to the separating hyperplane. In the case of nonlinear probes, we can treat the second layer of the MLP as a linear classifier and take the same view. This perspective associates a vector to each grid tile, corresponding to the classifier for that grid tile.

A natural question is whether these vectors display any geometric patterns. Figure 5 visualizes their configuration using PCA plots. To make the connection with board geometry clear, we have overlaid a grid in which the vector for a given grid tile is connected to the vectors that represent its neighbors. At left, as a baseline, are weights of probes trained on randomized GPTs; the result is a somewhat jumbled version of the board's grid. For classifier vectors, however, we see a somewhat clearer geometric correlation with the grid shape itself. This shape may reflect a set of correlations between neighboring grid cells, and could be an interesting object for future study. One point of interest is that probe geometry for the randomized network does not seem completely random. This may fit with the fact that linear probe baseline performance is better than chance, indicating some information about board state can be gleaned from random projections of game move tokens.

## B    ABLATION ON NONLINEAR PROBE ACCURACIES

Though high accuracies have been observed on nonlinear probes in section 3, we want to develop a deeper understanding on them. For example, we wish to understand when, during a game, a model has developed world representations of board tiles, where in Othello-GPT that information is stored, how difficult it is to decode that information, and when the model may forget that information.

As shown in Figure 6 (B), we plot probe accuracies of two-layer probes varying to two different experiment settings: (1) Probe Hidden Units: the number of hidden units $H$ in nonlinear probes; (2) Layer: at which layer the representations $x^l$ is taken out. With the increase of hidden units, i.e., probe capacity, probe accuracy is higher as it can capture more knowledge from the hidden space. For layer $l$, the accuracy peaks at midway, which is aligned with studies on natural language (Hewitt & Manning, 2019), where linguistic properties are found to be best probed in mid-layers. The format of our *what-how-where* plots is similar to the *what-when-where* plots in McGrath et al. (2021), except we ask how many hidden units within our nonlinear probe are necessary to achieve reasonable accuracy given each layer of the GPT instead of looking into number of training epochs.

We are also curious about when (in terms of game progression) these concepts are captured by Othello-GPT. Are these concepts updated immediately after each move? Will they persist or be

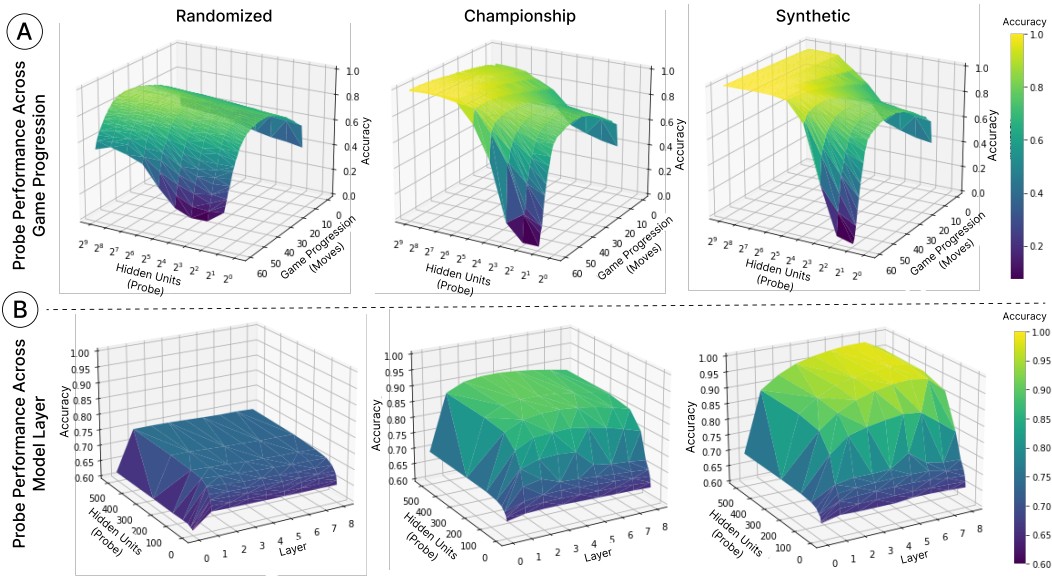

Figure 6: (A) *What-how-where* plots and (B) *What-how-when* plots of nonlinear probes trained on randomly-initialized GPT, on GPTs trained on championship and synthetic datasets respectively. (A) presents probe accuracy across an Othello game progression, while (B) presents accuracy across Othello-GPT layers.

forgotten with newer moves being made? To study this, we divide the data points for probe validation by how many steps the tile has been in its current state and plot *what* concept can be probed by *how* powerful probes *when* in the game progression in Figure 6 (A).

For nonlinear probes with a moderate number of hidden units, a parabolic accuracy curve is shown: concepts are best captured when they have existed for some time but not too long. This tells us: (1) forgetting happens when Othello-GPT changes its world representations; (2) there is a period of uncertainty before changes in board states are updated.

## C   PREDICTION HEATMAPS OF COUNTERFACTUAL BOARD STATES

In Figure 7, we can see one case of how intervention changes model prediction by intervening on the world representation of Othello-GPT. Note that the set of ground-truth legal moves are also changed by the intervention. In this case, both pre-intervention and post-intervention predictions have 0 errors. Figure 3 shows systematic results over $1,000$ cases. A random subset of 100 cases can be found in Supplementary Materials.

## D   INTER-PROBE INTERACTION

In Figure 8 we show the same case as in Appendix C of the world states probed from layers of Othello-GPT starting the 5th, which is the layer we start to intervene, $L_s$. We can observe that after intervention is successfully done on the 5th layer at C4, the flipped disc is corrected in the immediately following layer, as seen in the pre-intervention probing results at the 6-th layer. However, when the same intervention is done on later layers again, the model is more convinced that C4 should be black and stops to correct it. A total of 100 cases can be found in Supplementary Materials, corresponding to examples in Appendix C.

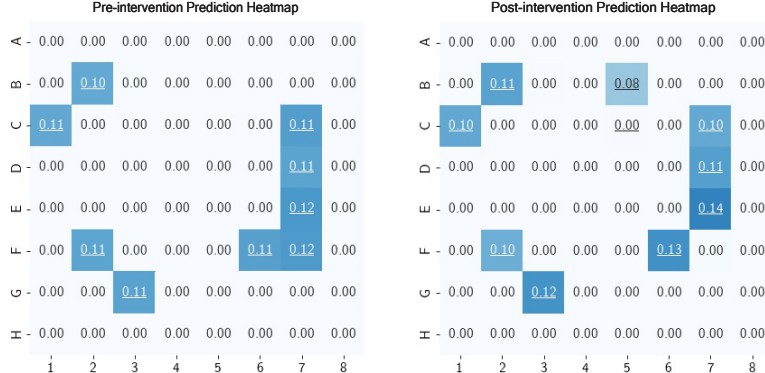

Figure 7: Heatmaps of the probability allocated to each tile on the board with nonlinear intervention done on Othello-GPT trained on synthetic dataset. To the left is the pre-intervention heatmap with pre-intervention legal moves underscored; to the right is post-intervention heatmap, with post-intervention legal moves underscored.

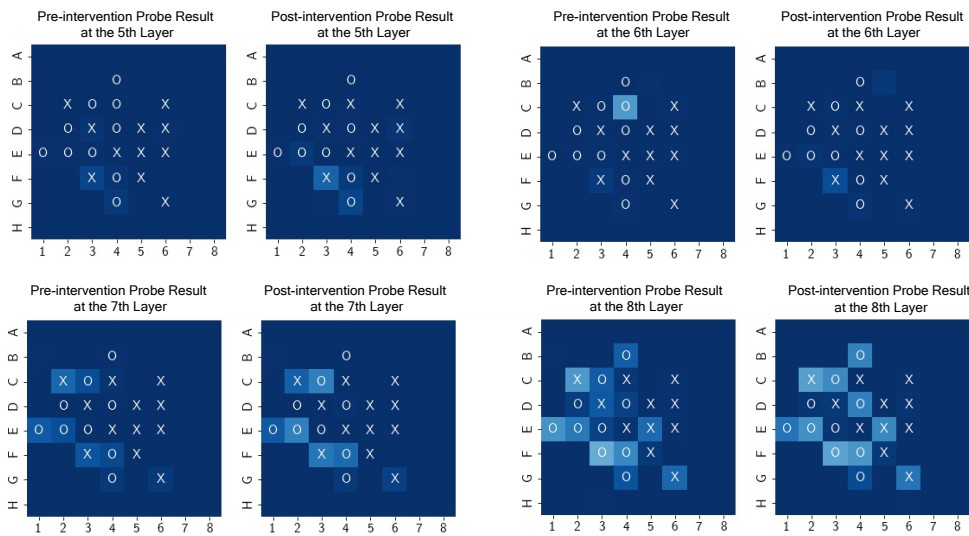

Figure 8: The world states predicted by probes pre-and-post-intervention on internal representations at different layers. Color encodes the confidence of the plotted top-1 prediction for each tile.

## E DISCUSSION OF LATENT SALIENCY MAPS

In the latent saliency maps created by attribution via intervention in Figure 4, when multiple discs are flipped, only the first one are found contributing, which is slighted misaligned to human understanding of the Othello rule. However, this is expected from the algorithm we are using because other flipped discs, even in the opposite states, still make the current prediction legal.

The attribution via intervention method succeeds at visualizing the AND-logic in the Othello rule to flip one straight line: there should be opponent discs in between *and* same-color disc at the other end. However, when the prediction flips more than one straight lines, intervening on one of them does not nullify the prediction. This is because the Othello rule is an OR-logic on top of the AND-logic: a move is legal when either one of the eight straight lines can be flipped. How to extract such knowledge with unlimited amount of intervention experiments at hand? We leave it for future research.

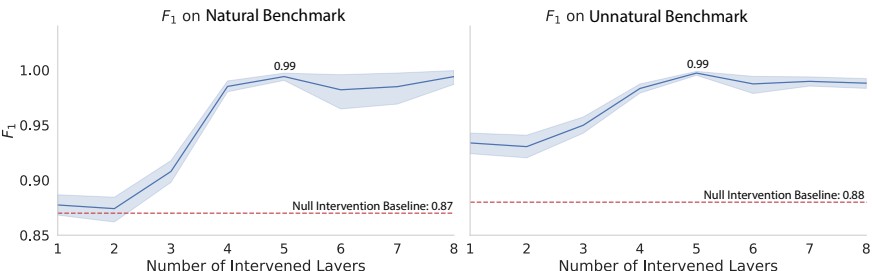

Figure 9: Same as Figure 3 except for reporting $F_1$ instead of Error.

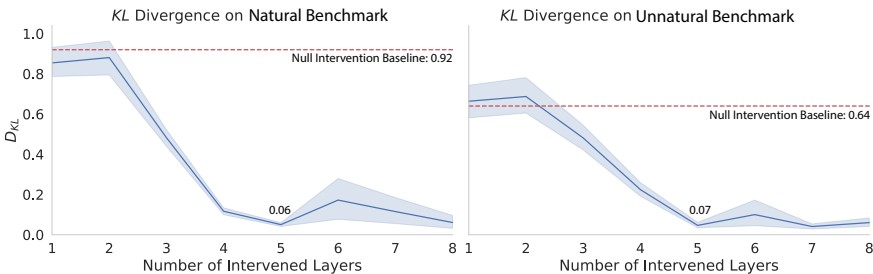

Figure 10: Same as above but for $D_{KL}$.

## F  ALTERNATIVE METRICS FOR THE INTERVENTION EXPERIMENT

Following the same multi-label classification framework that calculates the error in Figure 3, we report the $F_1$ score in Figure 9. We measure intervention success as a multi-class classification problem, rather than a top-1 prediction problem like in section 3, considering the fact that the later metric is almost saturated for the Othello-GPT trained on synthetic dataset and the intervention in benchmark only cause a set difference of 2.12 tiles on average.

Another line of thought to measure the alignment between model prediction and the intervened world representation is by measuring the KL divergence of the predicted next-move probability distribution from a discrete uniform prior distribution over all legal next-moves. Note that the baseline $F_1$ and $D_{KL}$ are the averages of the $F_1$ and $D_{KL}$ over all 1000 intervention cases, rather than treating the whole benchmark as one single multi-label classification problem or distribution distance calculation.

Error (Figure 3), $F_1$ (Figure 9), and KL divergence (Figure 10) show a similar trend, confirming that the proposed intervention technique is useful and validating the causality from world representations to model predictions.

## G  ABLATIONS ON INTERVENTION HYPER-PARAMETERS

Experiments find this optimization process is robust to different configurations of optimizer, learning rate $\alpha$, and number of steps.

The complete world state, including states of all the 64 board tiles, is encoded in a single internal representation $x$, while during intervention experiment, we only wish to change one of them. A natural question is: will the intervention operation flip tiles we are not not intending? It is possible but we can mitigate that by considering the cross entropy losses of other tiles as a regularization term, weighted by a hyper-parameter $\beta$. That is to say, the loss in main paper can be expanded as:

$$\mathcal{L}_{\text{CE}}(p_\theta(x), B') = \sum_{s \in B \neq B'} \mathcal{L}_{\text{CE}}(p_\theta^s(x), B'(s)) + \beta \sum_{s \in B = B'} \mathcal{L}_{\text{CE}}(p_\theta^s(x), B(s)).$$

We sweep $\beta$ at the best $L_s = 5$ on the natural benchmark and report average number of errors in Table 3. We can observe it does not clearly help.

Here we further discuss another hyper-parameter ablated in Figure 3, the starting layer for intervention, $L_s$. If we intervene with more than 5 layers, shallow layers which have not developed reliable world

| $\beta$ | 0.0 | 0.1 | 0.2 | 0.3 | 0.4 | 0.5 | 0.6 | 0.7 | 0.8 | 0.9 |
|---|---|---|---|---|---|---|---|---|---|---|
| error | 0.22 | 0.31 | 0.77 | 0.19 | 0.51 | 0.65 | 0.42 | 0.45 | 0.46 | 0.55 |

Table 3: Average number of errors under different $\beta$'s and $L_s = 5$ on the natural benchmark.

representations (according to Figure 6) are touched, making intervention hazardous. On the other hand, if we intervene only at the deepest layers, though the world representation can be intervened successfully (see Appendix D), the model does not have enough computation to adapt to the newer world representation and make predictions corresponding to it.

## H   STANDARD DEVIATION OF PROBE ACCURACY

In order to exclude the possibility that the differences in probe accuracies in Table 1 and Table 2 are caused by randomness, we run the same experiment for 100 times with different random seeds and report the standard deviations of them in Table 4 and Table 5, based on which we conclude that the probe accuracies are robust to randomness and there are significant differences between the probe accuracies of linear and nonlinear probes, and also between probes on randomized Othello-GPT and Othello-GPTs trained on different datasets.

| | $x^1$ | $x^2$ | $x^3$ | $x^4$ | $x^5$ | $x^6$ | $x^7$ | $x^8$ |
|---|---|---|---|---|---|---|---|---|
| Randomized | 0.04 | 0.04 | 0.05 | 0.06 | 0.06 | 0.06 | 0.09 | 0.06 |
| Championship | 0.05 | 0.04 | 0.06 | 0.07 | 0.06 | 0.06 | 0.06 | 0.06 |
| Synthetic | 0.06 | 0.06 | 0.07 | 0.06 | 0.05 | 0.04 | 0.06 | 0.08 |

Table 4: Standard deviations of error rates (%) of linear probes on randomized Othello-GPT and Othello-GPTs trained on different datasets across different layers ($x^i$ represents internal representations after the $i$-th layer).

| | $x^1$ | $x^2$ | $x^3$ | $x^4$ | $x^5$ | $x^6$ | $x^7$ | $x^8$ |
|---|---|---|---|---|---|---|---|---|
| Randomized | 0.05 | 0.06 | 0.07 | 0.04 | 0.04 | 0.03 | 0.04 | 0.03 |
| Championship | 0.05 | 0.09 | 0.08 | 0.06 | 0.05 | 0.05 | 0.10 | 0.06 |
| Synthetic | 0.07 | 0.09 | 0.04 | 0.06 | 0.06 | 0.09 | 0.05 | 0.05 |

Table 5: Standard deviations of error rates (%) of nonlinear probes on randomized Othello-GPT and Othello-GPTs trained on different datasets across different layers ($x^i$ represents internal representations after the $i$-th layer).

