# OpenReview forum: "Emergent World Representations: Exploring a Sequence Model Trained on a Synthetic Task"
_ICLR.cc/2023/Conference — ICLR 2023 notable top 5%_

### Official Review · Reviewer_1z26 · 2022-10-20

**Confidence:** 4
**Correctness:** 3
**Technical Novelty And Significance:** 2
**Empirical Novelty And Significance:** 3
**Recommendation:** 8

**Clarity, Quality, Novelty And Reproducibility:**

### Questions and concerns

1. In Section 2.2, the authors mention that the embedding layer consists of 60 vectors, where 60 is the vocabulary size. In an auto-regressive setting like GPT, I assume a set of special tokens (e.g., `<start of sequence>`, `<end of sequence>`, `<padding>`) would also be used? Especially in a batched training setting. Otherwise, how is the model trained?
2. In Section 2.3, the authors report that Othello-GPT's error rates are 0.01% and 5.17% when trained with the synthetic dataset and championship dataset, respectively. Is this difference only a matter of data size (20 million vs ~100k)? Have the authors tried to use a subset of the synthetic data with the same size as championship?
3. In Section 2.3, the author discuss about the skewed dataset, which truncates one of the four first moves. I am not fully convinced by that paragraph because if I understood correctly, the game tree will quickly cover those truncated states even if they are banned in the first move? I see footnote #3 but I still fail to fully understand, sorry.
4. In the interventional experiment, do the authors only consider changing between `{black, white}` and not `empty`? If so, could the authors provide more intuitions?
5. In Figure 2 caption, the authors mention that `we intervene at the temporally-last token`. Is this always the case or it's only true in this demonstrated example? If the latter, how does this work?
6. In the paragraph above Section 5, I fail to understand why `... comparing the prediction probability for each tile with 1/2N`. Why not `1/N`?
7. I like the latent saliency map idea. However, the examples shown in Figure 4 include many counterfactual game board states. What is the reason for looking at latent saliency maps on counterfactual data?


### Typos and minor issues
1. Section 2.2, `4rd` and `62st`.
2. In online instructions (e.g., [WikiHow](https://www.wikihow.com/Play-Othello)), it seems that in Othello, Black always goes first. I assume this is the case in the data the author collected. This rule is well aligned with Figure 1. However, in Section 2.3, the four possible opening moves seem wrong, they are valid tiles for white player.
3. Out of curiosity, do other types of LMs learn similar world models? For instance, BERT has an arguably different training procedure (fill-in-the-blank) as opposed to what GPT (next word prediction) uses.

**Strength And Weaknesses:**

### Strength

* **Nice task setting**: I like how the authors use board games as toy tasks for studying sequential data. Natural language, due to many confounding factors and properties, is not easy to be used for studies like in this work (e.g., flipping state of a tile). I'd love to see more work along this line, that can potentially help us to better model language, and the world that language describes.
* **Towards understanding LMs**: At the same time, this work propose concrete methods towards making LMs more interpretable and transparent.
* **Nice visualization method**: I like the latent saliency map method, which quite effectively represent dependencies among tiles. This can potentially be used in analyzing 1) word or concept dependencies in language generation; and 2) event or action dependencies in an offline RL setting (e.g., using decision transformer to model sequences of transitions, which essentially is an auto-regressive generation problem).
* **Writing**: I enjoy reading this paper, it's clear in most part of it.



### Weaknesses

* As the authors acknowledge, it remains unclear how methods proposed here can generalize to broader context, i.e., natural language, which is large language models designed for. I agree that this work is an "pilot study" towards that goal, but it would make the work much stronger if the authors can discuss more about, e.g., how board game trajectories connect with language, what are some common property, what are missing, and what are some potential paths towards that goal etc. As I mentioned among the strength, board games are good toy tasks studying sequential data, which gives researchers more control. It's just less clear in the sense of, how do we jump from this to language.
* Please refer to the questions and concerns below.

**Summary Of The Paper:**

In this work, the authors study if a GPT-based language model, trained on a set of board game move data, is capable of learning a "world model" of the game. To investigate this:
1. Given the representation generated by the LM, representing a game state, the authors try to use a probe (either a linear one or a non-linear one) to learn to predict the state of a tile (black/white/empty). They show that the non-linear probe can significantly outperform a baseline setting where board representations generated by random network is used. This suggests the game board state representation might be non-linear.
2. Then the authors conduct an interventional experiment. In which, given a game board $B$ and the probe's prediction (of a tile) conditioned on $B$'s representation, they intervene the representation so that the probe provides a flipped prediction (e.g., white --> black). In such case, if the LM (intervened) output desired possible legal moves, it would suggest that the learned representations had a causal effect on the model. Experiment results show that the intervened LM can indeed produce the desired prediction, rather than ones made from the original game board state $B$.
3. Extending the interventional experiment, the authors develop a visualization method, namely latent saliency maps. In which, for any tile on the board, the latent saliency map can show how much the network's prediction would change if an intervention had been applied to each of the other tiles.


**Summary Of The Review:**

In general, I like this topic, and I rate the current version of the paper around (slightly above) borderline. I look forward to reading the authors responses to my questions so I can better understand the work and thus have a more precise evaluation.

---

> ### Author Response · Authors · 2022-11-18
> **Response to Reviewer 1z26**
>
> We thank the reviewer for their valuable and constructive feedback. We have updated the document in response to their comments.
>
> **Subsetting experiments**
> In response to the reviewer’s helpful comments regarding training data size, we have trained an Othello-GPT on a random subset of the synthetic dataset of equivalent size to the championship dataset. The resulting error rate on the subsetted dataset is 16.16%, which is worse than training on either the championship or the entire dataset. The discrepancy does not impact our results, but does provide an interesting avenue for future research.
>
> **Skewed dataset**
> Our fundamental goal with the truncated-tree data set is to prevent memorization of input data. The network only sees sequences as input, not board states directly. We have updated the paper to reflect this more clearly.
>
> **Connecting back to natural language**
> Here is our response from the rebuttal summary:
> We have made changes in the introduction and conclusions of the paper to make explicit that we are using Othello, a constrained task, as a testbed for research on world representations, a key issue of interest in language modeling.
>
> **Intervention: Why do we only flip the tiles?**
> While we explored erasing a tile from black/white to blank, we did not include the experiment in the paper because of a peculiarity of Othello. At the game’s start, the center four tiles are always occupied. This means that the four probes for each center tile are never trained to predict an unoccupied space. Without examples of this in the training data, the interventions from black/white to blank will typically fail to either make the desired change to the targeted tile, or change surrounding tiles.
>
> **Counterfactual data and latent saliency maps**
> Given the reliability of the intervention technique introduced in Section 4, we can explore how the model responds to impossible or unnatural settings. This is the point of the counterfactual (unnatural) data: Given the prior set of moves, we can intervene to make an unnatural board state according to the game’s rules. Interestingly, we can now see how the model responds to this situation, which is guaranteed never to have been seen in the training data. Ultimately, it falls back on the learned representations to correctly choose how to move forward. We have edited the results section in the paper to emphasize this more clearly.
>
> **Intervention: only the temporally-last token?**
> We always intervene at the temporally-last token. We have updated the paper to clarify this point. The representations of the temporally-last token correspond to the board state of the whole game, on which the next-step predictions are based. In contrast, previous representations correspond to earlier board states. The interplay between the previous board states and next-step predictions is more complex and an avenue of future research.
>
> **Intervention metric in Figure 3**
> We have added additional evaluative metrics to support our conclusions. We included the prior metric in the Appendix and a discussion of its motivation. Here is our description of the metrics from the rebuttal summary:
>
> Given the confusion by several reviewers about our evaluation metrics, we moved our initial metric to the Appendix, replaced it in the paper with a more intuitive one (described below), and added another metric to the Appendix. The added experiments with the alternative metrics show similar results, which provides additional support for our thesis and conclusions.
>
> During interventions, we aim to measure how well the prediction is aligned with the world representation as a multi-label classification problem. Thus we need to make a prediction set from the probability distribution predicted by Othello-GPT. For the prior metric, we threshold the probability per tile by 1/2N. We have replaced this metric in the body of the paper with a more intuitive alternative, taking the top-N predictions as the prediction set and then reporting the error and $F_1$ score. In addition, we measure the alignment between model prediction and the world representation using the KL divergence of the predicted next-move probability distribution from a discrete uniform prior distribution over all legal next-moves. We included this metric in the Appendix.

---

> > ### Author Response · Authors · 2022-11-18
> > **Response to Reviewer 1z26 (continued)**
> >
> > **Typo in Game Example**
> > Our Othello implementation matches that of the Wikipedia article that the reviewer cites. Black does always go first. We have corrected the notation of the four possible opening moves to be valid for black rather than white.
> >
> > **Training**
> > Our Othello-GPT is a simplification of the original GPT. We also include the special token `<end_of_sentence>` to mark the end of generation. `<start_of_sentence>` is omitted as we always prompt Othello-GPT with a partial game. The final vocabulary in our implementation contains 61 words: excluding the four center board tiles, plus one special token.
> >
> > We have uploaded the code, data, and trained checkpoints at [this link](https://drive.google.com/file/d/1TT9bpGZTBSNYX26fGKd8HMk4iTYuqNwE/view?usp=share_link). The materials were missing from our submission due to the file size limitations of Open Review and anonymous GitHub repositories. We will make all materials freely available in our GitHub repository should the paper be accepted for publication.

---

> > ### Comment · Reviewer_1z26 · 2022-11-19
> > **Thank you**
> >
> > I have read other reviewers' comments as well as the authors' response. The response addressed and answered my concerns well, the additional information helped me to better understand the work.
> >
> > I would like to increase my score a bit, but since 7 is not an option this year, I will give an 8.

---

### Official Review · Reviewer_uxFu · 2022-10-25

**Confidence:** 3
**Correctness:** 3
**Technical Novelty And Significance:** 2
**Empirical Novelty And Significance:** 3
**Recommendation:** 6

**Clarity, Quality, Novelty And Reproducibility:**

Clarity - major issues with presentation of method and paper organization. See "Weaknesses" above. Additional clarifying questions below:
	- for Figure 3, what is the 95% CI computed over?
	- when generating the latent saliency map, are you using the latent representation from layer L = 5, since they give the lowest error on the intervention experiments? Or are you using some other layer(s)?

Quality - No major comments on quality, although I am not deeply familiar with the related probing literature which may have some bearing on the experiments presented here.

Novelty/originality - The main novelty here appears to be the application of these techniques to Othello. Similar probing, intervention, and feature saliency/attribution techniques have been used in other literature.

Reproducibility - limited. The authors have not shared the code, datasets, or trained model. While the championship dataset is taken from publicly available data, they do not share the random splits they use. It would therefore be difficult to reproduce these exact numbers. Details are provided for how to train Othello-GPT, but training details are not given for the probes.

**Strength And Weaknesses:**

Strengths
- Present new evidence that GPT variant can learn Othello board state, extending previous work on chess
- Presents some evidence that their intervention technique to alter the "emergent world state" is effective
- Using the intervention technique, they produce useful visualizations of "saliency" which describe how much a certain board position influences the prediction of Othello-GPT

Weaknesses
- There are several issues with clarity in the paper that made it difficult to understand on the first read.
	- I find the last paragraph of section 4 hard to understand, although I do think I grasped the method after reading the rest of the paper. It would help if the authors updated Figure 2 to show the actual metric/value that is used to determine if a representation is causal or not.
	- Similarly the description of the counterfactual case is not very clear. Please use notation to clarify the actual metric/value that is used to determine whether the internal representation is of the board "rather than just a sequence"
	- In the Related Work section, they authors refer to their work as having "a focus on the geometry of internal representations", but no geometry-related work is shown in the main results at all. This is only presented in the Appendix A, and this is not referred to anywhere in the main text.
	- In Tables 1 & 2, I believe "randomized" corresponds to probes trained on random weight GPT, rather than a random dataset. Table 1's caption makes it appear as if it is simply another dataset.
- The authors do not present any measures of variance for the probe error rates in Tables 1 & 2 (e.g. standard deviation or confidence interval over random seeds). This makes it difficult to interpret the difference between the probes trained with the actual Othello-GPT model and the ones trained with the random weight model.
- The authors also show that nonlinear probes perform better than linear probes, but do not address why this might be the case, despite the fact that they state this is a major contribution of the paper. What are we to take away from this result, other than "the probe may be recovering a nontrivial representation of board state"? The authors claim that the intervention experiments "validat[e] this hypothesis", but they do not discuss this hypothesis again. It appears as if this claim may be supported by Appendix A but they again do not refer to it at all.
- Finally, the authors fail to make any additional argument as to why these results would be compelling to a wider audience beyond those interested in games / Othello. In the conclusion the authors write that "the tools described in this paper—nonlinear probes, layerwise interventions, and latent saliency maps—may yet prove useful in natural language settings". However it's unclear to me how these specific tools would contribute anything above and beyond current work in NLP using probes, interventions and feature attribution.

**Summary Of The Paper:**

EDIT: Increased original score from reject to borderline accept after reading the responses and other reviews. I have also increased my empirical novelty/significance score from a 2 to a 3.

Inspired by previous work on language models (LMs) learning chess, this paper trains a GPT model to predict legal moves in the board game Othello, which they call Othello-GPT. They then aim to go beyond the work on chess to understand the internal representations of the model, which they call the "emergent world model". To accomplish this, they design simple probes to understand representations of the game board in the model. They then present an intervention technique to alter the internal representation of the board, and show evidence that this alters the predictions of a probe trained on the model's world state representations. Finally, they show a visualization that they call a latent saliency map that uses the intervention to visualize the contributions of different board positions to the system's predictions.

**Summary Of The Review:**

Overall, I do not think this paper is suitable for ICLR. The technical approaches themselves do not seem very novel, although my familiarity with this literature is not deep, and the Related Work discussion provided is quite brief. The main novelty of this work is in the application of these probing, intervention, and feature attribution techniques to the game Othello, which would likely only be of interest to a narrow audience. Finally, the issues with clarity and organization of the paper make this work appear rushed to submission.

---

> ### Author Response · Authors · 2022-11-18
> **Response to Reviewer uxFu**
>
> We thank the reviewer for the useful and constructive feedback. We have clarified several elements of the paper in our updated version.
>
> **Choice of Othello and connecting back to natural language settings**
> Here is our response to the rebuttal summary:
>
> We have made changes in the introduction and conclusions of the paper to make explicit that we are using Othello, a constrained task, as a testbed for research on world representations, a key issue of interest in language modeling.
>
> **Novelty**
> Whether or not GPT-style networks build internal world models is a matter of debate, so identifying them, in this case, has an interest independent from Othello. See our comments above. We also build on nonlinear probing techniques and contribute both intervention and attribution via intervention techniques, which have not been previously proposed. By leveraging these techniques, we can produce latent saliency maps, which are new to the literature.
>
> **Nonlinear and Linear Probes**
> There are two ways to interpret the reviewer’s questions regarding probing. First, “why isn’t there a linear representation (which could then be learned with a linear probe)?” Alternatively, they might ask, “Why does the probing matter since it is a relatively well-described technique?”
>
> Given the first interpretation, looking for a linear representation makes sense, as linear probes work in many other contexts. However, they do not work well in the Othello context, as evidenced by the reported error rates. We do not know why a nonlinear representation arises, but that would be an excellent thread of future research.
>
> Suppose the reviewer is concerned about the general emphasis on probing methods. In that case, we note that the motivation of this paper is not probing in isolation but rather to uncover latent, controllable world representations of the Othello board state. We note that the board state cannot be memorized from the input data but must be constructed by reconciling the underpinning processes that lead to gameplay. Appendix A provides a more visual way to see this latent representation but ultimately parallels our probing results.
>
> **Variance of probe error rates**
> To understand how much of the accuracy is caused by randomness, we retrained probes with multiple random seeds to confirm that, aligned with our initial experiments, the variances are below 0.1. This variance is statistically insignificant compared to probes trained on Othello-GPTs for randomized or different datasets, where the average difference is 5 and 25 for linear and nonlinear probes, respectively. We report the variances of probe accuracies in Table 4 and Table 5 in Appendix G.
>
> **Metrics Clarification**
> We have added additional evaluative metrics to support our conclusions. We included the prior metric in the Appendix and a discussion of its motivation. Here is our description of the metrics from the rebuttal summary:
>
> Given the confusion by several reviewers about our evaluation metrics, we moved our initial metric to the Appendix, replaced it in the paper with a more intuitive one (described below), and added another metric to the Appendix. The added experiments with the alternative metrics show similar results, which provides additional support for our thesis and conclusions.
>
> During interventions, we aim to measure how well the prediction is aligned with the world representation as a multi-label classification problem. Thus we need to make a prediction set from the probability distribution predicted by Othello-GPT. For the prior metric, we threshold the probability per tile by 1/2N. We have replaced this metric in the body of the paper with a more intuitive alternative, taking the top-N predictions as the prediction set and then reporting the error and $F_1$ score. In addition, we measure the alignment between model prediction and the world representation using the KL divergence of the predicted next-move probability distribution from a discrete uniform prior distribution over all legal next-moves. We included this metric in the Appendix.
>
> **Reproducibility**
> We have uploaded the code, data, and trained checkpoints at [this link](https://drive.google.com/file/d/1TT9bpGZTBSNYX26fGKd8HMk4iTYuqNwE/view?usp=share_link). The materials were missing from our submission due to the file size limitations of Open Review and anonymous GitHub repositories. We will make all materials freely available in our GitHub repository should the paper be accepted for publication.

---

> > ### Comment · Reviewer_uxFu · 2022-11-23
> > **Improved score**
> >
> > I thank both the authors and other reviewers for their responses. I do think the additional changes have improved the clarity and apparent significance of the paper, and will be increasing my score to a 6.

---

### Official Review · Reviewer_LWko · 2022-10-26

**Confidence:** 4
**Correctness:** 4
**Technical Novelty And Significance:** 3
**Empirical Novelty And Significance:** 3
**Recommendation:** 8

**Clarity, Quality, Novelty And Reproducibility:**

** Novelty & relevance **

While the paper looks at a relatively well-explored domain (transformers playing board games), I believe that a deeper dive into the subject makes the novelty of the paper high enough to be of interest to a large part of the ICLR community. The novelty is, overall, up to the standards of the ICLR conference.

Such in-depth studies of model behaviour in controlled domains are extremely important, as they shed light on architectures that are much harder to analyze in more "naturalistic" domains (such as NLP). I believe that such studies (when well executed) form a crucial component  of meaningful research progress in using these architectures, and are of high interest to a large portion of the ICLR community.

As a side note, while it is known that one can explicitly train the network to recount the game state in, say, a game of chess, an important difference in this case is that the network is not explicitly trained to do so, so we can see that the world state representation emerges naturally.

** Clarity **

The paper is well written and is a pleasure to read.

There are a few issues, however, most of which can be easily resolved:
"We discuss reasons for the difference between the error rates for the synthetic and championship
models later in the paper." - it is crucial to add an internal reference. I actually failed to find a discussion of this issue. Intuitively, it's because the dataset is much smaller, but I think it's worth a direct mention. Alternatively, the line can be removed and the reason for the difference can be assumed to be obvious.

Pages 7 and 8 of the paper were quite confusing. While the main message was clear, I struggled to understand the metrics used and I was confused by the last visualization. The algorithm did not help very much, since not all terms are clearly defined.

To be more precise, I think that the last visualization is not actually confusing, but appears to be confusing because of how it is introduced in the text. This part: "The basic idea is simple. For any tile on the board, we ask how much the network’s prediction would change if we applied the intervention of the previous section to change the state of that tile." I think it's best to rephrase it, since it does not mention that the authors focus on the top move prediction. Since the network prediction is generally a distribution over different moves, I expected that the change in move probability would be somehow aggregated over different tiles.

Lastly, and most importantly, I was confused by the paragraph associated with figure 3 ("To measure how well the prediction is aligned with ground-truth legal moves, we calculate a prediction set by comparing the prediction probability for each tile with 1/2N ..."). I don't fully understand why this metric was used, and why this threshold. It may be better to report the same metric as before (top move error rate), split by tile "circumstance". With two possible circumstances: 1) error rate in the case when the previous top move remained legal, 2) when it went from legal to illegal. Additionally, it may be worth looking at how often the network is switching to newly introduced legal moves.

It may be that the authors' metric already captures these ideas in an aggregated way. Nevertheless, I believe that an absolute minimum is to provide a much more thorough description and justification of the metric choice (potentially in the appendix). And I think that it's highly desirable to add additional metrics (similar to what I suggested above) to provide deeper insight. If the latter is impossible due to computational considerations -- it's not a catastrophe, but it does hurt the paper a little.

** Quality **

Generally, the paper states its research question and addresses it precisely. I thoroughly enjoyed its crisp, well-thought-out approach, and I believe that the quality of the paper overall is up to the highest standards.

There is one exception to that, however, which I described in the last two paragraphs of the "Clarity" section. It's not absolutely crucial, and is perhaps more an issue in reporting than in quality per se, but it does negatively affect the quality of the paper.

I deeply hope that it can be fixed before (and if) the paper is published.

** Reproducibility **

The contribution follows the highest standards of reproducibility.

** Other suggestions **

These suggestions may be of interest to the authors, but they did not affect my judgment and my evaluation is not contingent on authors addressing any of them.

Relevant literature:
The paper "Life after BERT: What do Other Muppets Understand about Language?" may be of interest as a systematic empirical evaluation of LMs understanding of language (https://aclanthology.org/2022.acl-long.227/)

It may also be reasonable to mention the relation of your work to works that explicitly train Transformers to incorporate arbitrary changes into their world state representation (e.g. https://arxiv.org/abs/2104.05500, which uses the game of life as one of the testbeds),

Experiment modification:
I find that it could be reasonable to either first train the model on the simulated data and then fine-tune in model on the championship dataset, or include a prefix or some other type of switch token to be able to prompt the model to either output "championship" or "simulated" moves and train on both datasets at once. This way, the model would be able to reliably learn to make legal moves, and, when prompted, will switch to a "competitive mode". Otherwise, the value of the championship mode is very much diminished, as the model clearly did not get enough data even to learn to make valid moves.

** Typos **

a 3-way categorical the probability distribution -> a 3-way categorical probability distribution

whether a nonlinear probes would -> whether nonlinear probes would

** UPD **: Edited minor typos in my review

**Strength And Weaknesses:**

Strengths:
- The paper addresses a highly relevant problem, and clearly states it.
- The experiments are well-planned and well-executed, directly addressing the research problem posed.
- The paper is clearly written, and is a pleasure to read.

Weaknesses:
- The choice of the metric used in the part 4.2 (EVIDENCE FOR A CAUSAL ROLE FOR THE REPRESENTATION) is not fully justified, and the calculation process is described too briefly, which may cause confusion.

**Summary Of The Paper:**

Using Othello game as the testbed for their research, the paper addresses a challenging question of whether transformers learn reasonable world-state representations, or simply exploit surface-level statistics of the data. First, the authors identify that the board state is recoverable from the network's representation, and then go an extra mile that this internal representation is causally connected with the network behavior.

**Summary Of The Review:**

I thoroughly enjoyed reading this paper. I think that the contribution clearly defined the problem it is focused on, presents a well-defined plan of attack, and offers a crisp execution. The problem itself is highly important and sufficiently novel.

While not perfect, I believe that in its present state, the contribution is above the threshold of acceptance, and can potentially become even better with relatively minor adjustments.

** UPD, after the rebuttal period **

I have read the authors' responses, which, I believe, address most of the concerns voiced by me and/or other reviewers (these concerns were predominantly minor in the first place).

I initially thought very highly of the paper, now I am even more confident in my assessment. I think it's a great paper, and I deeply hope that it gets accepted!

---

> ### Author Response · Authors · 2022-11-18
> **Response to Reviewer LWko**
>
> We thank the reviewer for their thoughtful and constructive feedback. We improved our writing, paying close attention to Sections 4 and 5.
>
> **Metrics in Section 4.2**
> We have added additional evaluative metrics to support our conclusions. We included the prior metric in the Appendix and a discussion of its motivation. Here is our description of the metrics from the rebuttal summary:
>
> Given the confusion by several reviewers about our evaluation metrics, we moved our initial metric to the Appendix, replaced it in the paper with a more intuitive one (described below), and added another metric to the Appendix. The added experiments with the alternative metrics show similar results, which provides additional support for our thesis and conclusions.
>
> During interventions, we aim to measure how well the prediction is aligned with the world representation as a multi-label classification problem. Thus we need to make a prediction set from the probability distribution predicted by Othello-GPT. For the prior metric, we threshold the probability per tile by 1/2N. We have replaced this metric in the body of the paper with a more intuitive alternative, taking the top-N predictions as the prediction set and then reporting the error and $F_1$ score. In addition, we measure the alignment between model prediction and the world representation using the KL divergence of the predicted next-move probability distribution from a discrete uniform prior distribution over all legal next-moves. We included this metric in the Appendix.
>
> **Notation**
> We updated Section 5 to include all definitions.
>
> **Introduction of Figure 4**
> We have rewritten the discussion of Figure 4 to better align with the content of the figure.
>
> **Internal Reference**
> As suggested by the reviewer, we have deleted the following sentence: "We discuss reasons for the difference between the error rates for the synthetic and championship models later in the paper."

---

> > ### Comment · Reviewer_LWko · 2022-11-23
> > **Thank you for your response.**
> >
> > I am glad to hear about the improvements. I initially thought very highly of the paper, and I am now even more confident in my assessment. Good luck, and I hope that your paper gets accepted!

---

### Official Review · Reviewer_VKpz · 2022-10-27

**Confidence:** 4
**Clarity, Quality, Novelty And Reproducibility:** please see above
**Correctness:** 4
**Technical Novelty And Significance:** 4
**Empirical Novelty And Significance:** 4
**Recommendation:** 8

**Strength And Weaknesses:**

Strength

The experiments are well motivated and well designed. The claims are well supported and convincing. The findings are significant.

The paper is extremely well written and crystal clear. I personally find it enjoyable to read.

Weakness

The training is a little strange to me. Why only predict y_T but not each y_t given its history? Just like how LM is pretrained.

There are some typos and grammatical errors.
E.g., parenthesis not matched for "(an example of which can be seen in Figure 2.", "as well as the change in next-step prediction in".
E.g., spacing is often weird, esp. before "Figure"s.

More visualizations may be helpful, e.g., when talking about the "counterfactual case".

**Summary Of The Paper:**

This paper studies whether LMs learn good internal representations that encode the actual process of generating the data, or simply memorizing surface features.
The paper investigates this problem in a game setting, i.e., Othello.
It presents a range of experiment results implying that, although the LM has no knowledge about the game rules, its learned representations encode information about the game board state.

**Summary Of The Review:**

This empirical paper carries out carefully designed and carefully executed experiments and have very interesting findings.
It has a lot of useful implications for future work.

This is the best submission that I have reviewed so far in 2022.

---

> ### Author Response · Authors · 2022-11-18
> **Response to Reviewer VKpz**
>
> We thank the reviewer for their positive and helpful feedback. We have fixed the typos and grammatical errors in the updated version of the paper.
>
> To the question on training: Our approach is essentially standard autoregressive LM training, e.g., similar to GPTs. At training time, for a game sequence of length $T$, there are $T-1$ partial games (each taken as input by the model to predict the next token), and therefore we calculate $T-1$ cross-entropy losses (one for each token except the first one). They are summed together as the final loss to be optimized. In Section 2.2, we only write about the loss for one such partial game, but in the implementation, all of them are used, so there is no difference between our training and standard GPT training.

---

### Author Response · Authors · 2022-11-18
**Summary of Changes**

We thank the reviewers for their thoughtful and constructive feedback. We respond to each reviewer’s points below but include a summary of key themes here:

**Motivation behind the metric choice and further evaluation comparing alternative metrics (Reviewer LWko, 1z26)**
Given the confusion by several reviewers about our evaluation metrics, we moved our initial metric to the Appendix, replaced it in the paper with a more intuitive one (described below), and added another metric to the Appendix. The added experiments with the alternative metrics show similar results, which provides additional support for our thesis and conclusions.

During interventions, we aim to measure how well the prediction is aligned with the world representation as a multi-label classification problem. Thus we need to make a prediction set from the probability distribution predicted by Othello-GPT. For the prior metric, we threshold the probability per tile by 1/2N. We have replaced this metric in the body of the paper with a more intuitive alternative, taking the top-N predictions as the prediction set and then reporting the error and $F_1$ score. In addition, we measure the alignment between model prediction and the world representation using the KL divergence of the predicted next-move probability distribution from a discrete uniform prior distribution over all legal next-moves. We included this metric in the Appendix.

**Relation of Othello to NLP tasks (Reviewers 1z26, LWko)**
We have made changes in the introduction and conclusions of the paper to make explicit that we are using Othello, a constrained task, as a testbed for research on world representations, a key issue of interest in language modeling.

**Lack of clarity on “counterfactual” board states (Reviewers uxFu, LWko, 1z26)**
We have changed the term to “unnatural” board state and explained how this benchmark is a stringent test of out-of-distribution data.

**Lack of clarity on interventional experiments (Reviewers 1z26, uxFu)**
We have reworded this section to make it easier to understand.

**Fixing minor typos (All)**
We have corrected all typos helpfully pointed out by the reviewers.

---

### Decision · Program_Chairs · 2023-01-20

**Decision:**

Accept: notable-top-5%

**Justification For Why Not Higher Score:**

The audience of this paper may be limited, as pointed by Reviewer uxFu.

**Justification For Why Not Lower Score:**

All the reviewers voted for acceptance. By checking the only weak-accept review, I did not find any major concerns. By reading the paper by myself, I agree with the reviewers that this paper is good and inspiring.

**Metareview: Summary, Strengths And Weaknesses:**

This paper studies whether LMs learn good internal representations that encode the actual process of generating the data, or simply memorizing surface features, with the Othello game as an example.  A list of analysis methods are conducted to study a LM learned to play Othello. Both the interventional experiments and the visualization provide useful taking-home messages.
As agreed by all the reviewers, the problem is novel and inspiring; the analysis approaches are well-designed; the claims are well supported and convincing. Moreover, the paper is well written and is a pleasure to read. As a result, all of us voted for acceptance.

An additional note: The only weak-accept review does not raise any major concerns. The reviewer agreed that this paper is good and inspiring, but the audience of this paper may be limited. I work in both RL and NLP, by reading the paper by myself, I believe this paper is inspiring to any people who likes to learn more about how the large LMs work, and the research can be potentially generalized to their research in the NLP domain.

**Note From Pc:**

if the above contains the word "oral" or "spotlight" please see: "oral" presentation means -> notable-top-5% and "spotlight" means -> notable-top-25%. As stated in our emails, we are disassociating presentation type from AC recommendations